# Genetic Diversity and Structure of Japanese Endemic Genus *Thujopsis* (Cupressaceae) Using EST-SSR Markers

**Michiko Inanaga [1], Yoichi Hasegawa [2], Kentaro Mishima [1]** 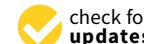 **and Katsuhiko Takata [3,*]**

1   Forest Tree Breeding Center, Forestry and Forest Products Research Institute,
    Forest Research and Management Organization, 3809-1 Ishi, Juo, Hitachi, Ibaraki 319-1301, Japan;
    inasuzume@affrc.go.jp (M.I.); mishimak@affrc.go.jp (K.M.)
2   Department of Forest Molecular Genetics and Biotechnology, Forestry and Forest Products Research
    Institute, Forest Research and Management Organization, 1 Matsunosato, Tsukuba, Ibaraki 305-8687, Japan;
    yohase@gmail.com
3   Institute of Wood Technology, Akita Prefectural University, 11-1 Kaieizaka, Noshiro, Akita 016-0876, Japan
*   Correspondence: katsu@iwt.akita-pu.ac.jp; Tel.: +81-185-52-6900

**Abstract:** The genus *Thujopsis* (Cupressaceae) comprises monoecious coniferous trees endemic to Japan. This genus includes two varieties: *Thujopsis dolabrata* (L.f.) Siebold et Zucc. var. *dolabrata* (southern variety, Td) and *Thujopsis dolabrata* (L.f.) Siebold et Zucc. var. *hondae* Makino (northern variety, Th). The aim of this study is to understand the phylogeographic and genetic population relationships of the genus *Thujopsis* for the conservation of genetic resources and future breeding. A total of 609 trees from 22 populations were sampled, including six populations from the Td distribution range and 16 populations from the Th distribution range. The genotyping results for 19 expressed sequence tag (EST)-based simple sequence repeat (SSR) markers, followed by a structure analysis, neighbor-joining tree creation, an analysis of molecular variance (AMOVA), and hierarchical *F* statistics, supported the existence of two genetic clusters related to the distribution regions of the Td and Th varieties. The two variants, Td and Th, could be defined by their provenance, in spite of the ambiguous morphological differences between the varieties. The distribution ranges of both variants, which have been defined from their morphology, was confirmed by genetic analysis. The Th populations exhibited relatively uniform genetic diversity, most likely because Th refugia in the glacial period were scattered throughout their current distribution area. On the other hand, there was a tendency for Td's genetic diversity to decrease from central to southern Honshu island. Notably, the structure analysis and neighbor-joining tree suggest the hybridization of the two varieties in the contact zone. More detailed studies of the genetic structure of Td are required in future analyses.

**Keywords:** *Thujopsis dolabrata*; EST-SSR markers; varieties; population structure

## 1. Introduction

Conifers are a dominant plant type found in the vast boreal forests of the North American and Eurasian continents. They represent an important forest resource in many countries because of their superior wood properties, including straighter trunks and stronger yet lighter wood compared to those of most angiosperms [1]. Generally, wild populations of trees are an important genetic resource and are required for breeding programs that aid in the selection of new plant varieties suitable for withstanding a range of environmental conditions, diseases, and future climate change [2–4]. A principal requirement for conserving forest genetic resources is maintaining the genetic diversity within and among the populations of a species [5]. Phylogeographic and population genetic studies using neutral molecular



markers provide a means of identifying in situ units and can be used to determine diversity level and distribution, gene flow routes, and major genetic disjunctions within the species [6]. Therefore, it is important to examine the genetic diversity and phylogeography of a wide range of wild populations, including in the context of conifer breeding.

*Thujopsis* (Cupressaceae) is a genus of monoecious coniferous trees with wind-mediated pollen- and seed-dispersal systems. It is endemic to Japan and is one of the basal lineages of Cupressoideae in the Cupressaceae phylogenetic tree [7]. This genus includes two varieties: *Thujopsis dolabrata* (L.f.) Siebold et Zucc. var. *dolabrata* (Td) and *Thujopsis dolabrata* (L.f.) Siebold et Zucc. var. *hondae* Makino (Th). Taxonomically, the former is regarded as the southern variety and the latter as the northern variety. Td, which features somewhat horned cones, is widely distributed in the southern region of the Japanese Archipelago, whereas Th, which is characterized by denser needles and rounder cones, is a race distributed in northern Honshu and the southern region of Hokkaido [8,9] (Figures 1a and A1). It is difficult to distinguish between these varieties using morphology alone, as their morphology tends to vary continuously. Therefore, in this study, both varieties will be classified according to their regional distribution. Because of the valuable properties of the wood, the genus *Thujopsis* is one of the most important tree species in Japanese forestry, and plantations of Th have been actively established in Aomori, Niigata, and Ishikawa Prefectures (see Figure 1b and Table 1) [10]. It is therefore essential to understand the phylogeographic and population genetic relationships of the genus *Thujopsis* for the conservation of genetic resources and future breeding.

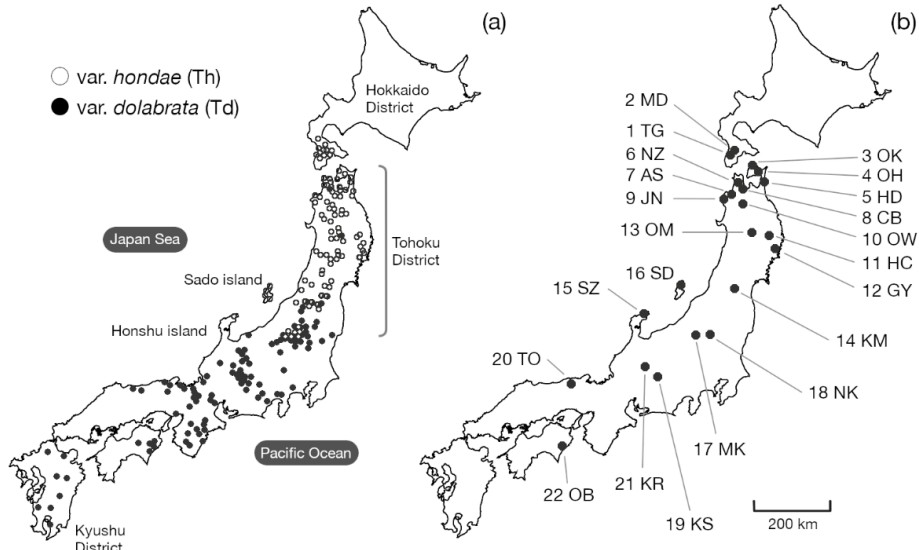

**Figure 1.** (**a**) Classical distribution range of genus *Thujopsis* defined by morphological differences between varieties shown in Kurata [9], var. *dolabrata* (Td, black dots) and var. *hondae* (Th, white dots). (**b**) Locations of the 22 sampled populations. Numbers correspond to the population numbers in Table 1.

Higuchi et al. [11] suggested that seven natural populations from the Th distribution region, as well as other Japanese conifers distributed over a wide area, showed a relatively low $F_{ST}$ (0.046). The two populations that were located in the marginal part of the Th distribution region and the five populations in the northern part of the distribution region were genetically different in the neighbor-joining tree and structure analysis [11]. Ikeda et al. [12] also analyzed the genetic structure of natural forests using populations distributed over almost the same geographical area as studied by Higuchi et al. [11]. They suggested that the distribution area of Th included four regional groups (Hokkaido and Aomori prefectures, Iwate and Yamagata prefectures, Niigata, and Ishikawa prefectures) of natural populations [12]. These findings are important for illustrating the genetic structure of Th;

however, they did not include Td populations. Therefore, a study examining both Td and Th is needed to determine the extent to which these two varieties are genetically distinct.

**Table 1.** Locations of *Thujopsis dolabrata*. Populations 1 to 16 fall within the range of var. *hondae* and 17 to 22 fall within var. *dolabrata*.

| No | Pop Name | Pop Code | Prefecture | Distribution Region | Latitude/Longitude | N |
|----|----------|----------|------------|---------------------|--------------------|---|
| 1 | Minamidate | MD | Hokkaido | var. *hondae* | 41.870072,140.283394 | 15 |
| 2 | Todogawa | TG | Hokkaido | var. *hondae* | 41.823216,140.208334 | 23 |
| 3 | Okoppe | OK | Aomori | var. *hondae* | 41.478333,140.952778 | 27 |
| 4 | Ohata | OH | Aomori | var. *hondae* | 41.386911,141.048666 | 30 |
| 5 | Higashidori | HD | Aomori | var. *hondae* | 41.085490,141.314990 | 30 |
| 6 | Nakazato | NZ | Aomori | var. *hondae* | 40.987595,140.456850 | 31 |
| 7 | Ajigasawa | AS | Aomori | var. *hondae* | 40.677765,140.184354 | 12 |
| 8 | Choubousan | CB | Aomori | var. *hondae* | 40.903131,140.605565 | 15 |
| 9 | Juniko | JN | Aomori | var. *hondae* | 40.558209,139.964216 | 30 |
| 10 | Owani | OW | Aomori | var. *hondae* | 40.448351,140.590625 | 22 |
| 11 | Omyoujin | OM | Iwate | var. *hondae* | 39.654310,140.896561 | 29 |
| 12 | Hayachine | HC | Iwate | var. *hondae* | 39.582572,141.481268 | 24 |
| 13 | Goyousan | GY | Iwate | var. *hondae* | 39.207849,141.716187 | 24 |
| 14 | Kaminoyama | KM | Yamagata | var. *hondae* | 38.083196,140.304773 | 19 |
| 15 | Sado | SD | Niigata | var. *hondae* | 38.215756,138.451880 | 31 |
| 16 | Suzu | SZ | Ishikawa | var. *hondae* | 37.402306,137.165028 | 17 |
| 17 | Minakami | MK | Gunma | var. *dolabrata* | 36.839137,138.972695 | 35 |
| 18 | Nikko | NK | Tochigi | var. *dolabrata* | 36.822614,139.440342 | 48 |
| 19 | Kiso | KS | Nagano | var. *dolabrata* | 35.727433,137.620934 | 32 |
| 20 | Kuraiyama | KR | Gifu | var. *dolabrata* | 35.985774,137.216638 | 35 |
| 21 | Toyo-oka | TO | Hyougo | var. *dolabrata* | 35.509750,134.637190 | 44 |
| 22 | Obitani | OB | Tokushima | var. *dolabrata* | 33.844595,134.327933 | 36 |

Population numbers (No); sample size of each population (N).

The aim of this study was to examine the genetic structure of the genus *Thujopsis* and to determine the relative contributions of the genetic structure between varieties. In the present study, simple sequence repeat (SSR) markers, which were representative neutral and co-dominant genetic markers, were used in order to provide basic information that would be useful in the conservation of *Thujopsis* genetic resources in natural forests.

## 2. Materials and Methods

### 2.1. Sampling and Study Sites

A total of 609 trees from 22 populations, including 6 populations from the Td distribution range and 16 populations from the Th distribution range, were sampled (Table 1; Figure 1b). Fresh needles were collected and stored at −30 °C. The sampled trees were each separated by more than 30 m. Sample sizes varied among the populations (Table 1). This variation reflects the sampling of large individuals (older trees) from some populations. Moreover, sampling was not attempted in dangerous areas where the topography was too steep.

### 2.2. DNA Extraction and Genotyping

Total genomic DNA was extracted from 100-mg needles using the hexadecyltrimethylammonium bromide (CTAB) method [13] with minor modifications. The genotypes of the sample trees were determined for 19 markers that were developed from expressed sequence tag (EST)-based simple sequence repeat (SSR) markers for the genus *Thujopsis* [14] (Table 2). In addition, universal primers were attached to each forward primer to efficiently incorporate fluorescent dyes during PCR for multiplexing [15]. The PCR was performed in a volume of 10.0 μL containing 10–120 ng of template DNA, 0.15 μM of forward primer, 0.5 μM reverse primer, 0.2 μM of either one of the tail primers fluorescently labeled by FAM, VIC, or NED, and 5.0 μL of Go Taq Master mix (Promega Corporation,

Madison, MI, USA). A GeneAmp PCR System 9700 thermal cycler (Applied Biosystems, Foster City, CA, USA) was used (Applied Biosystems, Foster City, CA, USA) with the following thermal profile: initial denaturation at 94 °C for 2 min, followed by 30 cycles of denaturation at 94 °C for 30 s; an annealing temperature of 60 °C for 30 s, and an extension at 72 °C for 30 s; then a final extension at 72 °C for 5 min. The amplified PCR products of each individual were classified into five groups according to the fragment size and the type of fluorescent marker (Table A1). Each group of PCR products was separated by capillary electrophoresis using a 3100 Genetic Analyzer (Applied Biosystems), and genotypes were scored with Geneious 7.0.4 software (Biomatters Ltd., Auckland, New Zealand).

**Table 2.** Genetic diversity measures estimated at 19 microsatellite loci.

| Variety | No | Pop Code | N | Allele Number | Allelic Richness | Private Allele | $H_o$ | $H_e$ | $F_{IS}$ |
|---|---|---|---|---|---|---|---|---|---|
| Th | 1 | MD | 15 | 6.30 | 5.86 | 1 | 0.646 | 0.639 | 0.025 |
|  | 2 | TG | 23 | 6.70 | 5.54 | 1 | 0.593 | 0.623 | 0.071 |
|  | 3 | OK | 27 | 7.50 | 5.97 | 2 | 0.673 | 0.662 | 0.002 |
|  | 4 | OH | 30 | 8.00 | 6.02 | 1 | 0.628 | 0.664 | 0.071 |
|  | 5 | HD | 30 | 7.80 | 5.93 | 2 | 0.646 | 0.641 | 0.010 |
|  | 6 | NZ | 31 | 8.70 | 6.55 | 0 | 0.689 | 0.682 | 0.005 |
|  | 7 | AS | 12 | 5.80 | 5.84 | 3 | 0.746 | 0.646 | −0.112 |
|  | 8 | CB | 15 | 6.50 | 6.12 | 0 | 0.662 | 0.660 | 0.032 |
|  | 9 | JN | 30 | 7.80 | 6.08 | 3 | 0.681 | 0.664 | −0.009 |
|  | 10 | OW | 22 | 6.80 | 5.76 | 1 | 0.624 | 0.636 | 0.042 |
|  | 11 | OM | 29 | 7.90 | 6.13 | 4 | 0.679 | 0.663 | −0.006 |
|  | 12 | HC | 24 | 6.80 | 5.56 | 1 | 0.618 | 0.603 | −0.004 |
|  | 13 | GY | 24 | 7.60 | 6.19 | 0 | 0.672 | 0.651 | −0.012 |
|  | 14 | KM | 19 | 6.70 | 5.87 | 3 | 0.637 | 0.624 | 0.006 |
|  | 15 | SD | 31 | 7.30 | 5.67 | 0 | 0.637 | 0.623 | −0.005 |
|  | 16 | SZ | 17 | 6.50 | 5.74 | 1 | 0.573 | 0.587 | 0.054 |
|  | Average |  |  | 7.17 | 5.93 | 1.4 | 0.650 | 0.642 |  |
| Td | 17 | MK | 35 | 7.20 | 5.61 | 1 | 0.609 | 0.611 | 0.018 |
|  | 18 | NK | 48 | 7.30 | 5.24 | 5 | 0.554 | 0.584 | 0.062 |
|  | 19 | KS | 32 | 6.40 | 5.15 | 1 | 0.554 | 0.561 | 0.027 |
|  | 20 | KR | 35 | 6.20 | 4.70 | 2 | 0.496 | 0.500 | 0.022 |
|  | 21 | TO | 44 | 5.90 | 4.68 | 1 | 0.542 | 0.548 | 0.023 |
|  | 22 | OB | 36 | 5.50 | 4.20 | 1 | 0.513 | 0.504 | −0.005 |
|  | Average |  |  | 6.42 | 4.93 | 1.8 | 0.545 | 0.551 |  |

*Thujopsis dolabrata* (L.f.) Siebold et Zucc. var. *hondae* Makino (Th); *T. dolabrata* (L.f.) Siebold et Zucc. var. *dolabrata* (Td); population numbers (No); sample size of each population (N); number of alleles per locus (allele number); allelic richness for standardized samples of 24 gene copies (allelic richness); number of private alleles per population (private allele); average observed heterozygosity ($H_o$); average expected heterozygosity ($H_e$); fixation index ($F_{IS}$).

*2.3. Data Analysis*

2.3.1. Genetic Diversity within Populations

The total number of detected alleles (TA), the observed heterozygosity ($H_o$), the total gene diversity ($H_T$), and Wright's inbreeding coefficient ($F_{IS}$) were calculated at each locus using FSTAT software version 2.9.4 [16].

The total number of detected alleles (allele number), the observed ($H_o$) and expected heterozygosity ($H_e$), and the number of private alleles were calculated for each population using GenAlEx 6.51b2 [17].

The allelic richness was standardized based on 12 individuals (24 gene copies), which was the smallest sample size among the populations. Fixation index values estimating inbreeding within individuals in a population ($F_{IS}$) were calculated. The significance of positive or negative values of $F_{IS}$ was tested based on 8360 randomizations with a Bonferroni correction. These calculations were performed using FSTAT [16]. The statistical independence of loci (linkage disequilibrium for all pairs of loci across populations) was also evaluated using FSTAT.

To compare genetic differentiation among populations and between loci, the relative genetic differentiation among populations defined under the infinite allele model (IAM; $F_{ST}$) [18] and the stepwise mutation model (SMM; $R_{ST}$) [19] were calculated using SPAGeDi version 1.5 [20]. The genetic differentiation coefficients ($G_{ST}$; analogous to $F_{ST}$) and a standardized measure, which had a range of

0–1 for all levels of genetic diversity ($G'_{ST}$), were calculated based on the allele frequencies for each locus using GenAlEx [21,22]. The significance of the deviations of $F_{ST}$, $R_{ST}$, $G_{ST}$ and $G'_{ST}$ (from zero) was evaluated by permutation tests.

The likelihood of a bottleneck within each population was examined using the two-phase model with 95% single-step mutations and 5% multi-step mutations with 1000 iterations, implemented in the program BOTTLENECK version 1.2.02 [23]. The one-tailed Wilcoxon test was used to detect an excess of expected heterozygosity ($H_e$) compared to that expected under mutation–drift equilibrium ($H_{EQ}$).

### 2.3.2. Genetic Structures among Populations and Distribution Regions

The presence of isolation-by-distance patterns in population differentiation was investigated by applying the Mantel test to the pairwise relationship between the geographic distances (transformed to natural logarithms) and genetic distances, $F_{ST}/(1 - F_{ST})$, between the populations according to Rousset's method [24]. Comparison analyses for Td population vs. Td population, Th population vs. Th population, and the species as a whole were performed, in order to separately examine the effect of isolation-by-distance within each distribution region and within the species. These calculations were performed using SPAGeDi version 1.5 [20].

For the analysis of molecular variance (AMOVA), we used Arlequin version 3.5.2.2 [25].

Hierarchical $F$ statistics were estimated using the R hierfstat package [26] ("varcomp.glob" and "boot.vc" functions), with the individuals ("Ind") nested within populations ("Pop"), nested within distribution regions ("Region"), and with 95% confidence intervals (CIs) from 1000 bootstraps. The significance of the hierarchical $F$ statistics was assessed using 10,000 permutations ("test.between" and "test.within" functions) [27]. These calculations were performed using R version 3.6.1 [28].

The genetic relationships among the populations were evaluated by constructing a neighbor-joining tree [29] using Poptree2 web [30,31]. Nei's chord distance ($D_a$) [32] was used to estimate the degree of genetic divergence of the populations. The node significances of the trees were evaluated using bootstrap probabilities based on 1000 replicates.

The Bayesian approach, which infers population structure and assigns individuals into clusters, was used, implemented in STRUCTURE version 2.3.4 software [33]. We performed 10 runs for each value of $K$ (number of putative populations) from 1 to 10, and employed the Markov chain method with 100,000 iterations (burn-in) and 100,000 Markov chain Monte Carlo repetitions. The simulation was performed under the admixture model with correlated allele frequencies (default parameters). Then the most appropriate cluster number ($K$) was selected using the criterion from Evanno et al. [34], which is based on $\Delta K$. To choose $K$ and identify sets of highly similar runs in multiple independent runs at a single $K$ value, we used CLUMPAK software [35].

## 3. Results

### 3.1. Genetic Diversity across All Populations

The total number of detected alleles for all populations at each locus ranged from five to 32, with an average value of 14.3 (Table A2). On average, over all loci, the observed heterozygosity ($H_o$) and gene diversity in the total population ($H_T$) were 0.621 and 0.691, respectively. The inbreeding coefficient ($F_{IS}$) ranged from −0.056 to 0.096. At all loci, the $F_{IS}$ values deviated significantly from zero.

The ranges of genetic diversity within the populations were 5.5 to 8.7 for allele number, 4.20 to 6.55 for allelic richness, 0.496 to 0.746 for $H_o$, and 0.500 to 0.682 for $H_e$ (Table 2). The average genetic diversities for Td and Th were 6.42 and 7.17 for allele number, 4.93 and 5.93 for allelic richness, 0.545 and 0.650 for $H_o$, and 0.551 and 0.642 for $H_e$, respectively. Private alleles were found in 18 populations, with a maximum value of five. There were four populations without private alleles. In all populations, observed heterozygosity was not significantly different from that expected for Hardy–Weinberg equilibrium. No evidence of significant linkage disequilibrium was detected in any of the total of 3420 tests for linkage disequilibrium between loci in the populations.

The genetic differentiation among population parameters $F_{ST}$, $R_{ST}$, $G_{ST}$ and $G'_{ST}$ varied among loci, ranging from 0.037 to 0.238, from 0.021 to 0.289, from 0.031 to 0.204, and from 0.098 to 0.502, respectively (Table A2). The genetic differentiation between populations over all loci ($F_{ST}$, $R_{ST}$, $G_{ST}$ and $G'_{ST}$) was 0.105, 0.096, 0.088, and 0.246, respectively. All these measures were significantly different from zero at all loci.

No evidence of a recent bottleneck was found (i.e., no $H_e$ excess compared to $H_{EQ}$) in the genotypes of all the populations.

### 3.2. Genetic Structure among Populations

All of the $F_{ST}/(1 - F_{ST})$ values between pairs of populations calculated for the three categories were significantly related to the natural logarithms of the geographic distances between them (Figure A2). For the Td vs. Td category, the intercept of the regression line (a) = −0.073 and the slope of the regression line (b) = 0.038, $R^2 = 0.567$ and $p < 0.01$. The Th vs. Th category showed a = −0.075, b = 0.022, $R^2 = 0.479$, $p < 0.000$. The species as a whole category showed a = −0.292, b = 0.071, $R^2 = 0.564$, $p < 0.000$.

The AMOVA suggested that the majority of the variation existed within population (Within Pop, 5.889, $p < 0.001$, Table 3). The results of the AMOVA and hierarchical $F$ statistics revealed a highly significant genetic divergence between the two distribution regions (Among Region, 0.606; $F_{Region/Total} = 0.088$; $p < 0.001$). The genetic differentiation was highly significant among populations, whereas the contribution of variety was relatively low between the two distribution regions (Among Pop, 0.393; $F_{Pop/Region} = 0.062$; $p < 0.001$).

**Table 3.** Hierarchical analysis of genetic structure using analysis of molecular variance (AMOVA) and hierarchical $F$ statistics.

| AMOVA | | | | Hierarchical $F$ Statistics | | | |
|---|---|---|---|---|---|---|---|
| **Source of Variation** | **Variance Components** | **Percentage of Variation** | | | **$F$** | | **95% CI** |
| Among Region | 0.606 | 8.80 | * | $F_{Region/Total}$ | 0.088 | * | 0.050–0.136 |
| Among Pop | 0.393 | 5.71 | * | $F_{Pop/Region}$ | 0.062 | * | 0.049–0.079 |
| Within Pop | 5.889 | 85.49 | * | $F_{Ind/Pop}$ | 0.018 | ns | 0.004–0.035 |

* $p < 0.001$.

The neighbor-joining tree based on $D_a$ distance reflected the geographical locations of the populations for the most part (Figure 2). However, genetic relationships between two distribution regions (Td and Th) and two intermediate populations (Minakami (MK) and Nikko (NK)) were not reliably supported because of low bootstrap values.

According to clustering analysis in STRUCTURE, the cluster number ($K$) was 2 (Figure A3). The distribution of populations for the two clusters was found to be geographically structured between the populations along the distribution regions (Figure 3). The proportion of cluster 1 was much higher in the Td distribution region, while that of cluster 2 was much higher in the Th region. Although the MK and NK populations were located in the Td region, individuals of these two populations belonged to both clusters 1 and 2.

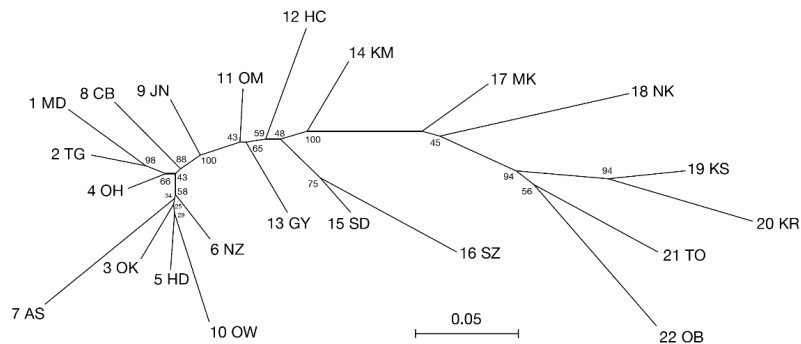

**Figure 2.** Neighbor-joining tree based on the $D_a$ distance of the 22 populations. The populations from No. 1 to 16 represent var. *hondae* (Th), whereas the populations from No. 17 to 22 represent var. *dolabrata* (Td).

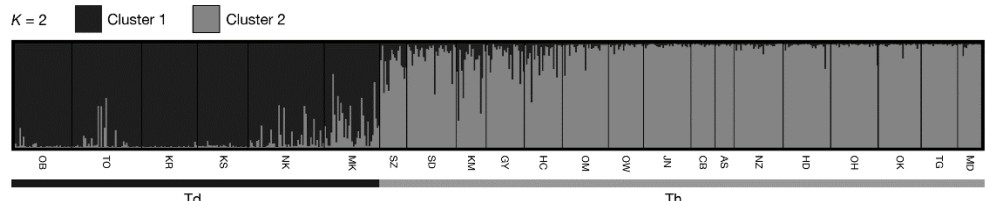

**Figure 3.** The proportions of cluster memberships at the individual level in the 22 genus *Thujopsis* populations based on STRUCTURE software analysis. Populations are represented from the left (southern populations) to the right (northern populations).

## 4. Discussion

### 4.1. Genetic Diversity at EST-SSR in Thujopsis

In the present study, 19 EST-SSR loci were used to estimate population genetic diversity and to investigate the genetic structure in 22 natural populations of the genus *Thujopsis*. The EST-SSR polymorphisms of the genus *Thujopsis* retained a nearly equal or slightly higher diversity compared to other Cupressaceae species (Table 2) [36–38]. In general, EST-SSR markers have a lower polymorphism than nuclear SSR markers [39,40]. Therefore, the EST-SSR marker loci used in the present study showed lower levels of polymorphism than the previous research on Th [11] and other conifers that are widely distributed in Japan [4,41,42].

A significant and relatively high value of overall population differentiation was found among the populations we examined (Table A2; $F_{ST} = 0.105$, $p < 0.001$; $R_{ST} = 0.096$, $p < 0.001$; $G_{ST} = 0.088$, $p < 0.001$; $G'_{ST} = 0.246$, $p < 0.001$). These values were clearly greater than those obtained in Ikeda et al. [12] ($F_{ST} = 0.039$; $G'_{ST} = 0.114$; EST-SSR). Katsuki et al. [43] reanalyzed and summarized the measures of population differentiation for major conifers distributed in the Japanese Archipelago, including *Cryptomeria japonica* ($F_{ST}$, 0.028; $R_{ST}$, 0.032; $G'_{ST}$, 0.125; SSR; [41]), *Chamaecyparis obtusa* ($F_{ST} = 0.039$; $G_{ST} = 0.040$; $G'_{ST} = 0.188$; SSR; [4]), *Picea alcoquiana* ($F_{ST} = 0.071$; $G'_{ST} = 0.164$; SSR; [44]), and *Picea jezoensis* ($F_{ST} = 0.101$; SSR; [45]). Additionally, Iwaizumi et al. [46] performed these measurements for *Pinus densiflora* populations distributed in Japan ($F_{ST} = 0.013$; $G_{ST} = 0.013$; $G'_{ST} = 0.122$; SSR). Compared with these species, we found a relatively moderate value for $F_{ST}$ and a high value for $G'_{ST}$ in the genus *Thujopsis*. In contrast to previous studies, which were conducted on a single species, measures of population differentiation in the genus *Thujopsis* may be higher because the study populations included two varieties (Td and Th). On the other hand, higher values for population differentiation measures have been observed in species, including isolated populations, of *Picea koyamae* ($F_{ST} = 0.209$; $R_{ST} = 0.173$; $G'_{ST} = 0.410$; SSR; [43]), *Sciadopitys verticillata* ($F_{ST} = 0.142$; SSR; [47]), *Abies mariesii* ($G_{ST} = 0.144$; allozyme; [48]), and *Pinus pumila* ($G_{ST} = 0.170$; allozyme; [49]). These species have narrow, isolated distributions that could reflect restricted gene flow

between populations because of habitat discontinuity [50]. Therefore, it is likely the case that the two populations were not completely genetically isolated, although we identified relatively high values for the population differentiation measures in the genus *Thujopsis*.

### 4.2. Comparison of Genetic Structure between Td and Th

The population relationships and genetic structures for the genus *Thujopsis* were analyzed, focusing on the relationship between the two variants. Structure analysis supported the existence of two genetic clusters related to the distribution regions (Figure 1a), i.e., the Td and Th varieties (Figure 3). These clusters were significantly differentiated based on AMOVA and hierarchical *F* statistics (Table 3). The neighbor-joining tree also supported these results, according to the high (100%) bootstrap probability of branches between the KM population (No. 14, Th) and the MK population (No. 17, Td) (Figure 2). Therefore, the two variants, Td and Th, could be defined by their provenance, in spite of the ambiguous morphological differences between these varieties.

The average values of allelic richness, $H_o$ and $H_e$, were relatively lower in Td than Th. Two factors may have contributed to the decline of genetic diversity in Td. First, demographic factors, such as postglacial colonization and a history of human overexploitation, could have played a role. If the refugia of the species were restricted to the southern region, a postglacial rapid expansion to northern regions would be expected to cause a series of founding events that would lead to a loss of alleles and homozygosity [51]. Similarly, tree species that have experienced population declines due to human overexploitation may show low genetic diversity and genetic bottlenecks [52–54]. However, no significant bottlenecks were detected in the population in the present study. This suggests that the low genetic diversity exhibited by each of the Td populations was probably caused by the natural characteristics of this variety, or other factors.

Since the Japanese Archipelago extends in a narrow arc from northeast to southwest, with the various mountain ranges probably acting as physical barriers, temperate plant species have generally migrated along the Pacific side, the Sea of Japan side, or the mountain slopes of the Archipelago. Thus, plants species migrated either southwards along the coasts or to lower altitudes into refugia during glacial periods, and expanded either northwards or to higher altitudes during interglacial periods [55]. In fact, many plant species distributed in Japan exhibit genetic divergence between the Pacific side and the Sea of Japan side (e.g., *Fagus crenata*, [56]; *Kalopanax septemlobus*, [36]). Additionally, as many tree species exhibit a long generation time, it is likely that not many generations have elapsed since the initial postglacial colonization. As a result, there has been less opportunity for genetic drift, and the large size of many plant populations could fossilize the genetic structure established at the time of colonization [57]. In the case of conifers, Kimura et al. [58] identified clear genetic divergence between two and four gene pools in *Cryptomeria japonica*. Two gene pools were distributed along the Sea of Japan side and along the Pacific Ocean side, while four gene pools suggested the potential of northern cryptic refugia and/or the potential of admixture events from several refugia between populations in the northern Tohoku district and an isolated gene pool on Yakushima Island (south of Kyushu district). As an example of fossilized genetic structures, Tsuda and Ide [59] suggested that the populations of *Betula maximowicziana* could be divided into a southern group (Central Honshu island) and a northern group (Hokkaido and Tohoku district) that originated from different refugia. They detected significant bottlenecks that may have been caused by processes of postglacial colonization and the species' characteristics and/or life history as a long-lived pioneer tree species. However, the present study indicates that the relatively high genetic differentiation of the genus *Thujopsis* did not fit these frequent patterns of genetic differentiation. The Td and Th varieties were distributed in the southern and northern parts of the Japanese Archipelago. No evidence of a genetic structure between the Sea of Japan side and Pacific Ocean side was found in this study. Th has similar diversity among populations, with relatively uniform values of allelic richness, $H_o$ and $H_e$. On the other hand, there was a tendency for allelic richness, $H_o$ and $H_e$, to decrease from the Minakami and Nikko groups to the Obitani group in Td. Structure analysis, and the locations of the Minakami and Nikko populations in the

neighbor-joining tree with low node significances, suggested that these two populations may contain Td and Th hybridization. In this case, the reason for the high genetic diversity in the Minakami and Nikko populations could be due to hybridization. The remaining four populations (Kiso, Kuraiyama, Toyo-oka, and Obitani) most likely represented pure Td populations, but we found no evidence of refugia in any of these.

Comparing the current distribution of Th with that of *Cr. japonica*, Aoyama [60] indicated that Th is more drought tolerant and can survive in colder climates. These physiological characteristics may have allowed Th to form refugia in southern Hokkaido and the areas below 500 m elevation in the Tohoku district during the last glacial period [60]. In the case of Th, refugia may have been scattered throughout the Tohoku district, which is the approximate current distribution range of Th. The relatively uniform values of allelic richness, $H_o$ and $H_e$, among the Th populations could be explained by this hypothesis. However, additional populations must be examined to fully understand the characteristics of Td.

*4.3. Contributions of the Breeding Program for the Genus Thujopsis*

Both varieties, Td and Th, are essential elements of natural forests in Japan, and logs from natural forests have historically been used for timber and other wooden materials. However, at present, Th is more important than Td in forestry. Plantations of Th are active mainly in Aomori, Niigata, and Ishikawa Prefectures (see Table 1) [10]. In Aomori, the demand for Th has recently increased. As a result, seed orchards were established in the beginning of 2003 [61]. In Niigata, plus trees on Sado Island were selected in 1989, and a seed orchard was established on Honshu island in 2009 [62]. Plantations in Ishikawa were established using rooted cuttings or saplings created by layering, and the majority of trees in these plantations are clones of 14 Th cutting/saplings [10]. In the Niigata and Ishikawa plantations, particular attention to interactions between natural populations through pollen flow and hybridization with Td is required. Th is found on Sado Island, and both Td and Th are distributed in the closest area on Honshu island (Figure 1). Similarly, the Suzu (SZ) population in Ishikawa, the present study site, is classified as Th; however, Td is distributed in neighboring areas. This study showed that these varieties are genetically different, and there is a risk that the seed orchard in Niigata may unintentionally produce seeds derived from hybridization between the variants. Furthermore, Th plantations in Ishikawa might risk introducing pollen of different variants into the surrounding natural Td forests. Since the breeding program of Th in Niigata and Ishikawa started relatively recently, the results of the present study could be useful for ongoing and future breeding programs, especially for the proper development and maintenance of seed orchards.

The results of the present study suggest that Td and Th can be distinguished by EST-SSR. As mentioned previously, distinguishing among varieties may be an important task in the future for breeding within the genus, *Thujopsis*. Several loci containing alleles with characteristic frequencies for Td and Th, respectively, were identified (Table A3). A combination of four loci, Tdest24, 39, 42, and 56, maintain a 100% bootstrap probability between varieties when constructing a neighbor-joining tree. Therefore, an analysis of these four loci is sufficient for the identification of the varieties.

## 5. Conclusions

Evidence from EST-SSR markers suggested that the two variants of the genus *Thujopsis*, *Thujopsis dolabrata* (L.f.) Siebold et Zucc. var. *dolabrata* (Td) and *Thujopsis dolabrata* (L.f.) Siebold et Zucc. var. *hondae* Makino (Th), were clearly distinct, as assessed using structure analysis and a neighbor-joining tree. Using these techniques, the two varieties could be defined by their provenance, in spite of the ambiguous morphological differences between them. The relatively uniform values of genetic diversity among the Th populations suggest that refugia of Th may have been scattered throughout the Tohoku district. On the other hand, there was a tendency for genetic diversity to decrease from central to southern Honshu island in Td. Structure analysis and the neighbor-joining

tree suggested hybridization in the contact zone between the two varieties. More detailed studies of the genetic structure of Td will be needed in the future.

**Author Contributions:** Study design and funding acquisition, K.M. and K.T.; needle sampling and fieldwork, M.I., Y.H., K.M., and K.T.; molecular and data analyses, M.I., and Y.H.; writing—original draft preparation, M.I.; writing—review and editing, M.I., Y.H., K.M., and K.T.; supervision and project administration, K.T. All authors have read and agreed to the published version of the manuscript.

**Funding:** This research received no external funding.

**Acknowledgments:** We would like to express our deepest gratitude to Miyako Sato, who was the first to summarize the research for this paper and provide draft results. Our heartfelt appreciation goes to Satomi Akiyama, whose enormous experimental support and encouraging words were invaluable. We would like to thank Seishiro Taki for his assistance in the fieldwork and for compiling information on sampling locations. Permission to use photographs of Td and Th used in Figure A1 in Appendix A was provided by the Forestry and Forest Products Research Institute (FFPRI) Database of Japanese Woods (http://db.ffpri.affrc.go.jp/WoodDB/JWDB-E/home.php). We are also indebted to the anonymous reviewers, who gave us invaluable comments.

**Conflicts of Interest:** The authors indicated no conflicts of interest.

## Appendix A

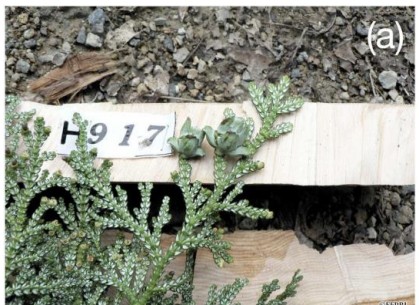 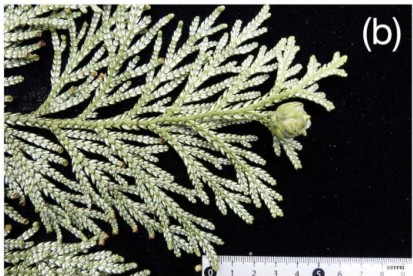

**Figure A1.** Typical morphology of (**a**) *Thujopsis dolabrata* (L.f.) Siebold et Zucc. var. *dolabrata* (Td) and (**b**) *Thujopsis dolabrata* (L.f.) Siebold et Zucc. var. *hondae* Makino (Th). Td (**a**) have horned cones, and Th (**b**) have denser needles and rounder cones. Photographs are cited from Forestry and Forest Products Research Institute (FFPRI) Database of Japanese Woods ((**a**) http://db.ffpri.affrc.go.jp/WoodDB/JWDB-E/detailA_coll.php?-action=browse&-recid=262420, and (**b**) http://db.ffpri.affrc.go.jp/WoodDB/JWDB-E/detailA_coll.php?-action=browse&-recid=265188).

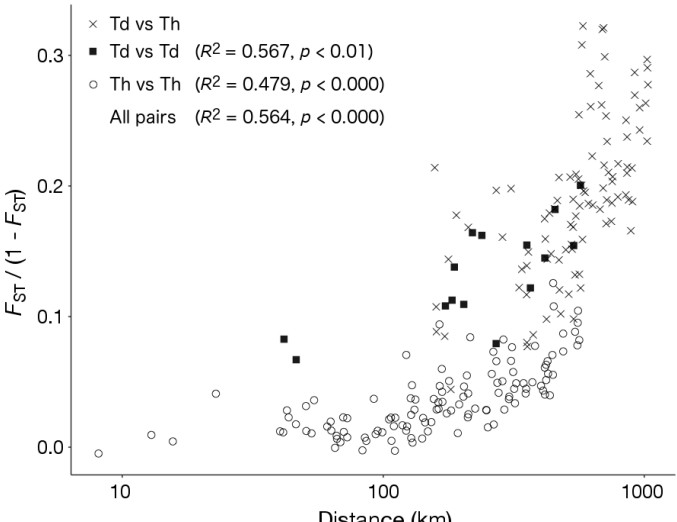

**Figure A2.** Relationships between pairwise genetic distances, $F_{ST}/(1 - F_{ST})$, and the geographic distance separating 22 *Thujopsis* populations.

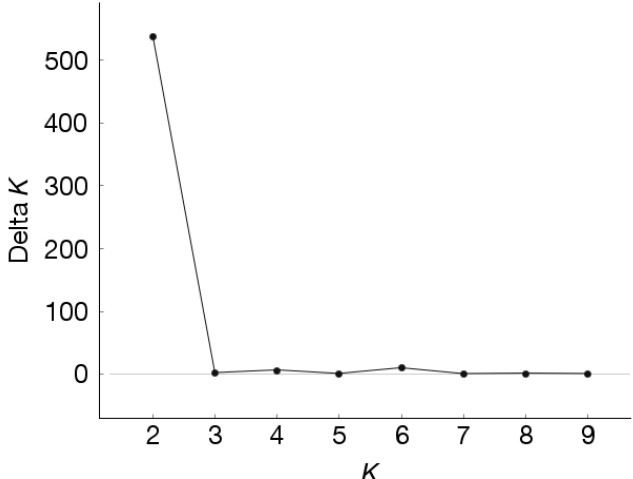

**Figure A3.** Relationships between the number of clusters (K) and the rate of the change in lnP(X|K) (Delta *K*), based on STRUCTURE analysis.

**Table A1.** Characteristics of 19 EST-SSR loci for use in the genus *Thujopsis*.

| Locus | Group | Repeat Motif | | Primer Sequence (5′–3′) | Allele Size Range (bp) |
|---|---|---|---|---|---|
| Tdest1 | 5 | (CT)11 | F: | GCCTCCCTCGCGCCATCAG GATTTTCTGACAGGCTTTGTTCTC | 137–173 |
| | | | R: | GTTTCTTAATTCCCAAGAGTGCTTATGAGTTC | |
| Tdest3 | 1 | (AT)11 | F: | GCCTCCCTCGCGCCATCAG CGGCCCAGGTTTCTGTACTC | 155–184 |
| | | | R: | GTTTCTTGCCCATTAAAGTCGGGTATTG | |
| Tdest11 | 3 | (AT)12 | F: | GCCTCCCTCGCGCCATCAGTGGGATACATACTGCATTTGTTAGG | 136–161 |
| | | | R: | GTTTCTTCTCCCCAAGCAAGTCACCAC | |
| Tdest14 | 4 | (AG)12 | F: | GCCTCCCTCGCGCCATCAGCAGTAGACAATTTCTGCAAATCACC | 152–190 |
| | | | R: | GTTTCTTTCCCTTTTGTTGGCATTATAGG | |
| Tdest17 | 3 | (AG)12 | F: | GCCTCCCTCGCGCCATCAGGCTTTTGATGTCCGCTATATCCTC | 160–176 |
| | | | R: | GTTTCTTGGAGATTCCAATGTTTGTCATGC | |
| Tdest21 | 3 | (AG)13 | F: | GCCTCCCTCGCGCCATCAGGTCCATCCATTCTCACTCCAAAG | 228–292 |
| | | | R: | GTTTCTTAGCAGACCCTATTTCACAGCATC | |
| Tdest24 | 4 | (AT)15 | F: | GCCTCCCTCGCGCCATCAGATACCATACAGCTTTCAGCCAG | 239–266 |
| | | | R: | GTTTCTTGCAGAACAAACGAATCAATGAGAG | |
| Tdest29 | 3 | (AC)16 | F: | GCCTCCCTCGCGCCATCAGAAACGACTCTGCTGGATTTCAC | 215–243 |
| | | | R: | GTTTCTTTTCCGCTCTTGATTTTCTCTCC | |
| Tdest35 | 2 | (CT)15 | F: | GCCTCCCTCGCGCCATCAGAAGCTATTGACCCTTCTCAGGATAC | 191–227 |
| | | | R: | GTTTCTTCCATGTTGAATTGTTCCCTTTC | |
| Tdest37 | 5 | (ATC)9 | F: | GCCTCCCTCGCGCCATCAGCCAAGCGACAGAAAACCATTC | 158–175 |
| | | | R: | GTTTCTTTCAGTCTCTTCCTCCTCCTCCTC | |
| Tdest38 | 1 | (ACC)9 | F: | GCCTCCCTCGCGCCATCAGTGACCATTCCTCCTCCTCCTC | 114–137 |
| | | | R: | GTTTCTTCATGTTTGCAGTTGAGAGAAGACC | |
| Tdest39 | 1 | (GCT)9 | F: | GCCTCCCTCGCGCCATCAGGCAGCACAGGAGAAGAAAGATG | 153–186 |
| | | | R: | GTTTCTTACAACAGCCACAACGTGTCC | |
| Tdest42 | 1 | (ACC)9 | F: | GCCTCCCTCGCGCCATCAGCTCCCTATCCCAACACCAACAC | 225–258 |
| | | | R: | GTTTCTTTGCCTACCTATCCTTCTTCTTCTCC | |
| Tdest43 | 2 | (CGG)9 | F: | GCCTCCCTCGCGCCATCAGGGTCCAATGCAGGTAATACAAGAAG | 134–167 |
| | | | R: | GTTTCTTTCCCCGCCAAGATACTCAAC | |
| Tdest45 | 5 | (GGT)12 | F: | GCCTCCCTCGCGCCATCAGTGAGGGTGGTGAGACAATTC | 208–235 |
| | | | R: | GTTTCTTCAAGATTTGGAACTCCTGCAAC | |
| Tdest49 | 2 | (GAT)10 | F: | GCCTCCCTCGCGCCATCAGGTGCCCTCAAAGTTACAGCAGTC | 221–248 |
| | | | R: | GTTTCTTGCAATCACCTCATCCTCACTTC | |
| Tdest53 | 4 | (CTT)13 | F: | GCCTCCCTCGCGCCATCAGCCAAAGCCCTTCCAGTAACATC | 241–305 |
| | | | R: | GTTTCTTGATGGAATGAGTGAATCTCAGGAAC | |
| Tdset56 | 2 | (AAG)9 | F: | GCCTCCCTCGCGCCATCAGCATTGCCCTTTGGAATATAGGATC | 142–167 |
| | | | R: | GTTTCTTGTTGCCCATCTGCTCTTCTTC | |
| Tdest58 | 4 | (AAG)13 | F: | GCCTCCCTCGCGCCATCAGCTGAACGGCGCCCTAATCTC | 151–188 |
| | | | R: | GTTTCTTGCCCACTCCTCAAATCCAAC | |

Groups of loci that were mixed when amplified PCR products of individuals were separated by capillary electrophoresis (Group).

**Table A2.** Genetic diversity measures estimated at 19 microsatellite loci.

| Locus | TA | $H_o$ | $H_T$ | $F_{IS}$ | | $F_{ST}$ | | $R_{ST}$ | | $G_{ST}$ | | $G'_{ST}$ | |
|---|---|---|---|---|---|---|---|---|---|---|---|---|---|
| Tdest1 | 19 | 0.889 | 0.904 | −0.037 | * | 0.065 | ** | 0.097 | ** | 0.052 | ** | 0.380 | ** |
| Tdest3 | 13 | 0.503 | 0.606 | 0.064 | ** | 0.135 | ** | 0.058 | ** | 0.113 | ** | 0.251 | ** |
| Tdest11 | 14 | 0.789 | 0.873 | 0.032 | ** | 0.078 | ** | 0.289 | ** | 0.066 | ** | 0.371 | ** |
| Tdest14 | 20 | 0.864 | 0.904 | 0.014 | * | 0.037 | ** | 0.024 | * | 0.031 | ** | 0.260 | ** |
| Tdest17 | 10 | 0.624 | 0.781 | 0.054 | ** | 0.175 | ** | 0.069 | ** | 0.157 | ** | 0.474 | ** |
| Tdest21 | 32 | 0.891 | 0.927 | 0.004 | ** | 0.041 | ** | 0.021 | * | 0.035 | ** | 0.349 | ** |
| Tdest24 | 16 | 0.710 | 0.767 | 0.023 | ** | 0.054 | ** | 0.205 | ** | 0.053 | ** | 0.199 | ** |
| Tdest29 | 8 | 0.358 | 0.409 | 0.025 | ** | 0.118 | ** | 0.153 | ** | 0.101 | ** | 0.163 | ** |
| Tdest35 | 32 | 0.852 | 0.935 | 0.040 | ** | 0.059 | ** | 0.095 | ** | 0.051 | ** | 0.470 | ** |
| Tdest37 | 5 | 0.398 | 0.484 | 0.018 | ** | 0.207 | ** | 0.205 | ** | 0.164 | ** | 0.281 | ** |
| Tdest38 | 10 | 0.655 | 0.692 | −0.034 | ** | 0.108 | ** | 0.104 | ** | 0.085 | ** | 0.239 | ** |
| Tdest39 | 7 | 0.456 | 0.490 | −0.056 | ** | 0.143 | ** | 0.173 | ** | 0.118 | ** | 0.212 | ** |
| Tdest42 | 13 | 0.565 | 0.732 | 0.031 | ** | 0.238 | ** | 0.054 | ** | 0.204 | ** | 0.502 | ** |
| Tdest43 | 14 | 0.714 | 0.753 | −0.011 | ** | 0.077 | ** | 0.121 | ** | 0.062 | ** | 0.218 | ** |
| Tdest45 | 9 | 0.462 | 0.581 | 0.096 | ** | 0.144 | ** | 0.136 | ** | 0.120 | ** | 0.252 | ** |
| Tdest49 | 8 | 0.228 | 0.272 | 0.051 | ** | 0.121 | ** | 0.125 | ** | 0.116 | ** | 0.155 | ** |
| Tdest53 | 21 | 0.863 | 0.888 | −0.025 | ** | 0.064 | ** | 0.099 | ** | 0.052 | ** | 0.342 | ** |
| Tdest56 | 8 | 0.626 | 0.730 | −0.010 | ** | 0.183 | ** | 0.082 | ** | 0.152 | ** | 0.411 | ** |
| Tdest58 | 12 | 0.361 | 0.402 | 0.044 | ** | 0.065 | ** | 0.068 | ** | 0.060 | ** | 0.098 | ** |
| average | 14.3 | 0.621 | 0.691 | 0.014 | ** | 0.105 | ** | 0.096 | ** | 0.088 | ** | 0.246 | ** |

Total number of detected alleles (TA); average observed heterozygosity ($H_o$); total gene diversity ($H_T$); Wright's inbreeding coefficient ($F_{IS}$) and significant deviations from Hardy–Weinberg expectations were tested; Weir and Cockerham's $F_{ST}$ ($F_{ST}$); relative genetic differentiation among populations defined under the stepwise mutation model ($R_{ST}$); genetic differentiation coefficient ($G_{ST}$); standardized measure of relative genetic differentiation among populations ($G'_{ST}$); *, $p < 0.01$; **, $p < 0.001$; $p$ values indicate the significance of deviations of $F_{IS}$, $F_{ST}$, $R_{ST}$, $G_{ST}$, and $G'_{ST}$ from zero, evaluated by permutation tests.

**Table A3.** Recommended set of markers for distinguishing Td and Th and their allele characteristics.

| | **Allele Frequencies across All Populations** | | | | |
|---|---|---|---|---|---|
| Locus | Allele Size | Td | Th | Td/Th | Th/Td |
| Tdest24 | N | 230 | 379 | | |
| | 239 | 0.152 | 0.003 | 57.6 | 0.0 |
| | 242 | 0.000 | 0.001 | | |
| | 243 | 0.030 | 0.000 | | |
| | 245 | 0.217 | 0.090 | 2.4 | 0.4 |
| | 247 | 0.424 | 0.429 | 1.0 | 1.0 |
| | 249 | 0.072 | 0.110 | 0.7 | 1.5 |
| | 251 | 0.057 | 0.173 | 0.3 | 3.1 |
| | 253 | 0.004 | 0.074 | 0.1 | 17.0 |
| | 255 | 0.002 | 0.051 | 0.0 | 23.7 |
| | 256 | 0.002 | 0.008 | 0.3 | 3.6 |
| | 257 | 0.026 | 0.028 | 0.9 | 1.1 |
| | 259 | 0.002 | 0.003 | 0.8 | 1.2 |
| | 261 | 0.009 | 0.004 | 2.2 | 0.5 |
| | 263 | 0.002 | 0.022 | 0.1 | 10.3 |
| | 265 | 0.000 | 0.004 | | |
| | 266 | 0.000 | 0.001 | | |
| Tdest39 | N | 230 | 379 | | |
| | 153 | 0.952 | 0.575 | 1.7 | 0.6 |
| | 156 | 0.000 | 0.001 | | |
| | 159 | 0.039 | 0.243 | 0.2 | 6.2 |
| | 162 | 0.000 | 0.003 | | |
| | 165 | 0.009 | 0.170 | 0.1 | 19.6 |
| | 168 | 0.000 | 0.003 | | |
| | 186 | 0.000 | 0.005 | | |

**Table A3.** *Cont.*

| Allele Frequencies across All Populations | | | | | |
|---|---|---|---|---|---|
| Locus | Allele Size | Td | Th | Td/Th | Th/Td |
| Tdest42 | N | 230 | 379 | | |
| | 225 | 0.028 | 0.001 | 21.4 | 0.0 |
| | 228 | 0.000 | 0.003 | | |
| | 231 | 0.004 | 0.018 | 0.2 | 4.2 |
| | 234 | 0.022 | 0.000 | | |
| | 237 | 0.041 | 0.215 | 0.2 | 5.2 |
| | 240 | 0.015 | 0.001 | 11.5 | 0.1 |
| | 243 | 0.498 | 0.042 | 11.8 | 0.1 |
| | 244 | 0.102 | 0.000 | | |
| | 246 | 0.048 | 0.600 | 0.1 | 12.5 |
| | 249 | 0.185 | 0.062 | 3.0 | 0.3 |
| | 252 | 0.007 | 0.041 | 0.2 | 6.3 |
| | 255 | 0.050 | 0.007 | 7.6 | 0.1 |
| | 258 | 0.000 | 0.009 | | |
| Tdest56 | N | 230 | 379 | | |
| | 142 | 0.002 | 0.000 | | |
| | 147 | 0.000 | 0.001 | | |
| | 152 | 0.370 | 0.080 | 4.6 | 0.2 |
| | 153 | 0.011 | 0.000 | | |
| | 155 | 0.022 | 0.463 | 0.0 | 21.3 |
| | 161 | 0.478 | 0.230 | 2.1 | 0.5 |
| | 164 | 0.117 | 0.223 | 0.5 | 1.9 |
| | 167 | 0.000 | 0.003 | | |

Sample size (N); *T. dolabrata* (L.f.) Siebold et Zucc. var. *dolabrata* (Td); *Thujopsis dolabrata* (L.f.) Siebold et Zucc. var. *hondae* Makino (Th); ratio of allele frequency, Td divided by Th (Td/Th); ratio of allele frequency, Th divided by Td (Th/Td).

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
