# Peer review of "Genetic Diversity and Structure of Japanese Endemic Genus Thujopsis (Cupressaceae) Using EST-SSR Markers"

_forests, doi:10.3390/f11090935_

Round 1

Reviewer 1 Report

The manuscript is well-presented, and significant work on many Thujopsis populations was done. The manusript fits the Scope of Forests. However, before publishing, it requires modification and extension.

The general impression is the lack of information about significance of work for breeding programs. I feel that Discussion can be significantly improved providing the overview of possible usage of author's findings for breeding programs.

As I see, there is also a lack of evident practical application of the results that the authors obtained. Is it possible to recommend this set of markers for distinguishing Td and Th despite their morphological similarity? If yes, which set of primers would you recommend and which pattern of fragments will be characterictic for each variety?

I also suggest to add the photos of varieties pointing to their morphological differences. Such Figure could be cited around lines 50-53.

Other suggested modifications are given by Lines:

Line 31 – varietas is a Latin word. I guess it is a misprint.

Lines 79-81 – The explanation is needed: why sample sizes are so different? How many plants were in each population? According to the Figure 5, sample size was quite different – it requires comments, and also a link to Table 1 would fit here, because it contains number of trees sampled.

Lines 86-87 – The table with primer sequences is needed, maybe as a supplementary file.

Lines 96-97 – More explanation about the 5 groups is needed. It is a key part of the analysis.

Line 108 – Also some clarification would be good: it not clear, which individuals and why 12.

Line 154 – The name of paragraph is a bit confusing.

Lines 176-181 – For this kind of analysis, coefficients of correlation are expected rather than coefficients of determination

Figure 4 must be explained better, including its legend. 

Table 1 - Italics is missed for species name

Table 2 also requires additional comments: p-values for which parameters and in comparison with what are given there?

Table 3 – As a suggestion, instead of dubbling population full names from the Table 1, the sample size could be given here. Lines "Average" are better to be with the bold font. 

Table 4 requires explanation. Does the highest percent of variation among individuals basically show that the chosen set of EST-SSR markers is not efficient in this case?

Lines 240-246 - It would be better to mention type of markers from cited articles.

Line 262 – Figure 5 instead of Figure 4?

Reviewer 2 Report

Dear Authors!
Your work seemed interesting to me, but I am not sure that it will be of interest to a wide range of readers. I will definitely report this conclusion to the editors.
In general, I can give a half-hearted response to your article. Among similar articles, it looks good. The merit of your work is the careful execution of experiments, high-quality statistical analysis of data and adequate conclusions.
However, I have some comments on the style of writing this article. I advise you to fix all the places where you write in the first person (we, our etc).
line 16: "we sought  to understand..."
lines 17-18 and 79: "We  sampled a total of 609 trees..."
line 73: "We used microsatellite markers..."
line 92: "We used a GeneAmp PCR System..."
lines 113-114: "we calculated  the relative genetic differentiation among populations..."
line 120: "We examined the likelihood..."
line 129: "We performed analyses..."
etc
line 250: "our  measures  of  population differentiation..."
line 251: "because our study populations included two varieties..."
etc
I also recommend moving bulky tables into supplemental materials so that pages 6-9 do not consist of them only. The quality of the drawings should be improved.
